# On Closed-Form Couplings

**Tobias Höppe**[1,2,3*]   **Stefan Bauer**[1,2,3]   **Qiang Liu**[7]   **Andrea Dittadi**[1,2,3†]   **Kirill Neklyudov**[4,5,6†]

[1] Technical University of Munich    [2] Helmholtz AI    [3] Munich Center for Machine Learning
[4] Mila – Quebec AI Institute    [5] Université de Montréal    [6] Institut Courtois
[7] University of Texas at Austin

## Abstract

Few-step generative modelling is an open challenge for flow models. Rectified flows tackle it by distilling a pre-trained "teacher" into a few-step "student", using strong noise–data couplings supplied by the teacher. For a finite dataset and a Gaussian probability path, the probability-flow vector field induced by the empirical distribution is available in closed form, which would allow us to skip training a teacher model. Surprisingly, these couplings turn out to be poor teachers and significantly reduce the performance of the student. We analyse this phenomenon empirically and theoretically, arguing that it stems from intrinsic ambiguity in the induced couplings caused by the strong sensitivity of terminal states to small initialisation perturbations. Under symmetry assumptions, we further prove that the closed-form probability-flow vector field preserves dataset symmetries and induces invariant Voronoi partitions.

## 1 Introduction

Diffusion and flow-based generative models learn a time-dependent vector field that transports samples from a simple base distribution to the data distribution by integrating an ODE (Song et al., 2021; Lipman et al., 2023; Sohl-Dickstein et al., 2015). Flow Matching (FM) trains a parametric field by regressing to the conditional velocity of a chosen probability path, enabling simulation-free training. In this view, the training target depends on how endpoint samples are paired along the path.

Within this paradigm, rectified flows and related straightening procedures improve sampling by leveraging couplings between noise and data (Liu et al., 2023; Lee et al., 2024; Kim et al., 2025). Rectified flows learn near-straight trajectories between endpoints by repeated rectification. This process uses matched noise–sample pairs obtained from a pre-trained model, which effectively provides a strong coupling between noise and data. These observations naturally suggest an appealing question: if rectification benefits from "good" couplings, could we avoid pretrained couplings entirely by using closed-form couplings? Concretely, given a Gaussian probability path and a dataset defining the target marginal, one can derive an "empirical" optimal regression target (vector field) for FM/diffusion models (Bertrand et al., 2025; Scarvelis et al., 2025). This could imply a shortcut for rectification.

However, the empirical vector field does not specify a one-to-one pairing between data points and noise. The induced ODE dynamics are strongly contracting as $t \to 0$, so entire regions of the prior are driven into the same datapoint when integrating towards $t = 0$, forming semi-discrete couplings (Mousavi-Hosseini et al., 2025). As a result, the closed-form coupling is inherently many-to-one and therefore does not yield a canonical choice of $(\mathbf{x}_0, \mathbf{x}_1)$ training pairs (Figure 4). In this work, we show that training on analytically constructed couplings fails in practice, not because the underlying objective is ill-posed, but because the choice of coupling is ambiguous and study an approach to mitigate this ambiguity.

Our contributions are manifold: (i) We characterise the coupling ambiguity that arises when instantiating closed-form dynamics as endpoint pairs, and empirically show a failure mode on CIFAR-10.

---

*Correspondence: `tobias.hoeppe@tum.de`.
†Equal advising.

(ii) We study alternative ways to construct usable pairings from closed-form dynamics: (a) SVGD-based couplings, and (b) a deterministic initialisation scheme, *PF-ODE Mean*, that removes the stochastic ambiguity at $t = \varepsilon$. (iii) We provide theoretical results describing the behaviour of the closed-form probability-flow vector field $\mathbf{v}^{PF}$ under symmetry assumptions on the data. In particular, we give sufficient conditions under which $\mathbf{v}^{PF}$ preserves dataset symmetries and induces invariant Voronoi partitions.

## 2   BACKGROUND

Let $\mathbf{x}_0 \sim p_0$ denote the data distribution on $\mathbb{R}^d$. We consider the Gaussian conditional path

$$p_t(\mathbf{x}_t \mid \mathbf{x}_0) = \mathcal{N}\big(\mathbf{x}_t \mid \alpha_t \mathbf{x}_0, \sigma_t^2 \mathbf{I}\big) \qquad t \in [0,1], \tag{1}$$

for differentiable schedules $\alpha_t, \sigma_t \in \mathbb{R}^*$ which induce marginals $p_t(\mathbf{x}) = \int p_t(\mathbf{x} \mid \mathbf{x}_0) p_0(\mathbf{x}_0) d\mathbf{x}_0$. There exists a deterministic probability-flow ODE $\frac{d\mathbf{x}_t}{dt} = \mathbf{v}_t(\mathbf{x}_t)$ whose solution has those marginals $p_t$. For Equation (1), the conditional velocity field is

$$\mathbf{v}_t(\mathbf{x}_t \mid \mathbf{x}_0) = \frac{\mathrm{d}\alpha_t}{\mathrm{d}t}\mathbf{x}_0 + \frac{\mathrm{d}\log\sigma_t}{\mathrm{d}t}(\mathbf{x}_t - \alpha_t\mathbf{x}_0). \tag{2}$$

The marginal (probability-flow) vector field is then the posterior average $\mathbf{v}_t(\mathbf{x}) = \int \mathbf{v}_t(\mathbf{x} \mid \mathbf{x}_0)\, p_t(\mathbf{x}_0 \mid \mathbf{x})\, d\mathbf{x}_0$. For an empirical distribution $\hat{p}_0(\mathbf{x}) = \sum_i^N \delta(\mathbf{x} - \mathbf{x}_0^i)$, this yields the closed-form field

$$\mathbf{v}_t^{PF}(\mathbf{x}) = \sum_{i=1}^N w_t^i(\mathbf{x})\Big(\frac{\mathrm{d}\alpha_t}{\mathrm{d}t}\mathbf{x}_0^i + \frac{\mathrm{d}\log\sigma_t}{\mathrm{d}t}\big(\mathbf{x} - \alpha_t\mathbf{x}_0^i\big)\Big) \qquad w_t^i(\mathbf{x}) = \frac{\mathcal{N}(\mathbf{x} \mid \alpha_t\mathbf{x}_0^i, \sigma_t^2\mathbf{I})}{\sum_{j=1}^N \mathcal{N}(\mathbf{x} \mid \alpha_t\mathbf{x}_0^j, \sigma_t^2\mathbf{I})}. \tag{3}$$

Since $\mathbf{v}_t^{PF}$ is induced by the empirical data distribution, the *PF-ODE* is strongly contracting as $t \to 0$, so integrating the exact field backwards maps any initial condition to one of the data points. In practice, Flow Matching approximates $\mathbf{v}_t^{PF}$ with a parametric model $\mathbf{v}^{FM}(\mathbf{x}, t)$ by least-squares regression to a Monte Carlo draw of the conditional velocity. Specifically, sample $\mathbf{z} \sim \mathcal{N}(\mathbf{0}, \mathbf{I})$, set $\mathbf{x}_t = \alpha_t\mathbf{x}_0 + \sigma_t\mathbf{z}$, and use the target $\frac{d\mathbf{x}_t}{dt} = \frac{\mathrm{d}\alpha_t}{\mathrm{d}t}\mathbf{x}_0 + \frac{\mathrm{d}\sigma_t}{\mathrm{d}t}\mathbf{z}$. Theoretically, a model with sufficient capacity would eventually converge to Equation (3).

## 3   CLOSED-FORM *PF-ODE* COUPLINGS UNDERPERFORM PRETRAINED COUPLINGS

We motivate our study by presenting unexpected results on CIFAR-10 when comparing $\mathbf{v}^{PF}$ couplings to couplings obtained from $\mathbf{v}^{FM}$. For a controlled comparison and throughout this paper, we construct *forward* couplings rather than the *backward* couplings commonly used in rectified flow training (unless otherwise stated). We do not use the backward construction for two reasons. First, closed-form couplings do not provide a mechanism for generating additional data, so we fix the number of couplings to the dataset size $N$. Second, integrating $\mathbf{v}^{PF}$ backwards does not guarantee a uniform assignment over data points since large regions of the prior may collapse to the same endpoint, while other data points receive no trajectories.

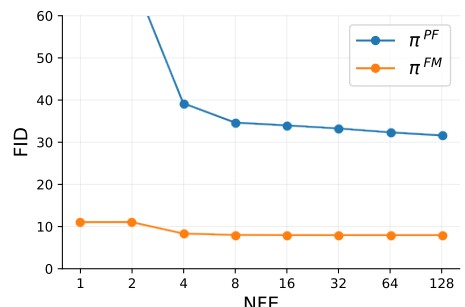

Figure 1: **Closed-form vs. pretrained couplings on CIFAR-10.** Note: for a fair comparison, we only use 50k couplings.

Concretely, for each data point $\mathbf{x}_0^i \sim p_0$ we sample $\mathbf{x}_\epsilon^i \sim \mathcal{N}\big(\alpha_\epsilon \mathbf{x}_0^i, \sigma_\epsilon^2\big)$ and integrate the chosen dynamics ($\mathbf{v}^{PF}$ or $\mathbf{v}^{FM}$) forward from $t = \epsilon$ to $t = 1$ to obtain a noise endpoint $\mathbf{x}_1^i$. This yields the coupling set $\pi^{(\cdot)} = \{(\mathbf{x}_0^i, \mathbf{x}_1^i)\}_{i=1}^N$, with $\pi^{PF}$ denoting

---

*We assume $\sigma_t > 0$ for $t \in (0,1]$, with $\alpha_0 = 1$, $\sigma_0 = 0$ and $\alpha_1 = 0$, $\sigma_1 = 1$.

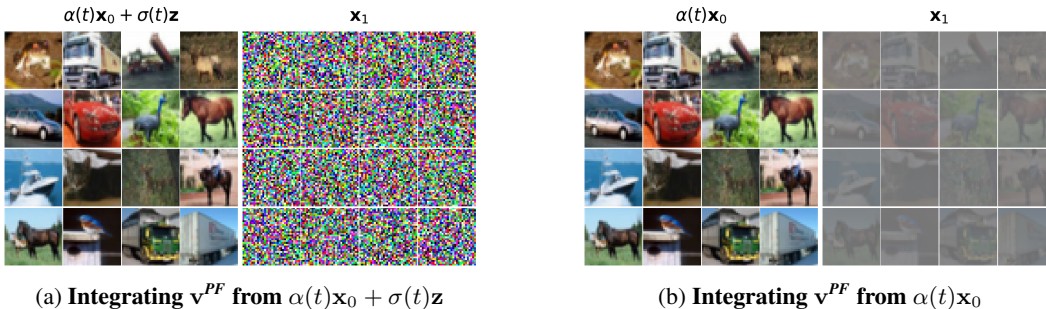

   (a) **Integrating $\mathbf{v}^{PF}$ from $\alpha(t)\mathbf{x}_0 + \sigma(t)\mathbf{z}$**          (b) **Integrating $\mathbf{v}^{PF}$ from $\alpha(t)\mathbf{x}_0$**

Figure 2: **Difference between *PF-ODE* and *PF-ODE Mean***. The difference in the initial condition is minor, since $t = 10^{-4}$ and is visually not identifiable. However, when sampling from $p_t$, our integrated points resemble Gaussian noise, while starting from the mean reduces the intensity of images but does not destroy the structure.

the set of closed-form couplings and $\pi^{FM}$ from a pre-trained model. Starting at $t = \epsilon$ avoids the singularity at $t = 0$. For training, we then follow the setup from Lee et al. (2024). Experimental details are given in Appendix B. We obtain reasonable FID scores when training on $\pi^{FM}$, whereas training on $\pi^{PF}$ performs markedly worse (Figure 1). This opens two questions: How do pretrained couplings differ from closed-form *PF-ODE* couplings, and can we find any closed-form couplings which yield good solutions?

**Pretrained couplings are consistent with *PF-ODE*.** To understand how the couplings differ, we first explore if the pretrained couplings do align with any possible analytical coupling obtained from $\mathbf{v}^{PF}$. For this, we first integrate a sample $\mathbf{x}_\epsilon^i$ from a datapoint $\mathbf{x}_0^i$ forward in time with $\mathbf{v}^{FM}$ to obtain $\mathbf{x}_1^i$ and then integrate $\mathbf{x}_1^i$ backwards in time with $\mathbf{v}^{PF}$. If we end up at $\mathbf{x}_0^i$, this proves that $\mathbf{v}^{FM}$ does indeed learn one of the couplings possible under $\mathbf{v}^{PF}$. Running this experiment on CIFAR-10 achieves a matching accuracy of 99.5 %, raising the question: if $\mathbf{v}^{FM}$ learns one possible coupling of $\mathbf{v}^{PF}$, why does training on $\pi^{PF}$ perform so much worse than $\pi^{FM}$. Before investigating this further, we perform an additional experiment with a different one-to-one coupling scheme, demonstrating that analytical couplings can perform well when the pairing is chosen from a smooth vector field.

**Smooth one-to-one couplings via Stein Variational Gradient Descent (*SVGD*).** As a choice for unambiguous couplings, we construct $\pi^{SVGD}$ using Stein Variational Gradient Descent (*SVGD*) (Liu & Wang, 2016). *SVGD* deterministically transports a set of particles towards a target distribution by a kernelised functional gradient step that combines attraction to high-density regions (via $\nabla_\mathbf{x} \log p(\mathbf{x})$) with a repulsive interaction term induced by a positive definite kernel. Concretely, given particles $\{\mathbf{x}^i\}_{i=1}^N$ and stepsize $\eta > 0$, SVGD iterates $\mathbf{x}^i \leftarrow \mathbf{x}^i + \eta\,\phi(\mathbf{x}^i)$ with

$$\phi(\mathbf{x}) = \frac{1}{N}\sum_{j=1}^N \Big[ k\big(\mathbf{x}^j, \mathbf{x}\big)\,\nabla_{\mathbf{x}^j} \log p\big(\mathbf{x}^j\big) + \nabla_{\mathbf{x}^j} k\big(\mathbf{x}^j, \mathbf{x}\big) \Big]. \tag{4}$$

Since SVGD is known to degrade in high dimensions (Ba et al., 2021), we use a 2D spiral dataset (Figure 5 B) to isolate coupling effects. In this setting, the resulting couplings yield substantially better training behaviour than $\pi^{PF}$ (Figure 3), and support few-step generation.

### 3.1   *PF-ODE Mean*: Deterministic initialisation removes ambiguity

Having verified that $\pi^{FM}$ is realisable under the closed-form probability-flow $\mathbf{v}^{PF}$, and that an unambiguous analytical coupling can be effective, we return to the *PF-ODE* couplings. A key difficulty is the instability of the *PF-ODE* near $t = 0$. Small perturbations of the initial condition can lead to large differences in the terminal state $\mathbf{x}_1$, which makes the association of a unique $\mathbf{x}_1$ to a given datapoint $\mathbf{x}_0$ ambiguous. To remove this ambiguity, we replace the stochastic initialisation with its conditional mean and use this as a deterministic starting point, which we call *PF-ODE Mean*. Concretely, instead of starting our integration from $\mathbf{x}_\epsilon = \alpha_\epsilon \mathbf{x}_0 + \sigma_\epsilon \mathbf{z}$, we start from $\mathbf{x}_\epsilon = \alpha_\epsilon \mathbf{x}_0$. Interestingly, we found that integrating $\mathbf{v}^{PF}$ forward from these initial states does not yield a Gaussian

distribution at $t = 1$ but only scales and warps the data structure (Figure 5 D). We have investigated this phenomenon on the CIFAR-10 dataset as well and found the same result (Figure 2). Since the terminal distribution is not Gaussian, we cannot use *PF-ODE Mean* to construct couplings for training. Nevertheless, the induced structure is unexpected, and we therefore investigate when it can be characterised and exploited.

**Symmetry preservation under *PF-ODE Mean*.** Motivated by highly structured samples produced by *PF-ODE Mean*, we next ask if such behaviour can be characterised more explicitly. In particular, we investigate whether *PF-ODE Mean* exhibits any form of structure preservation, and if so, for which classes of data distributions this can be established theoretically. Under certain symmetry assumptions on the data, we can characterise this behaviour and show that *PF-ODE Mean* preserves the induced symmetries (Proposition 3.1, with detailed proof in Appendix C).

**Proposition 3.1.** *For each nonzero datapoint* $\mathbf{x}_0 \in \{\mathbf{x}_0^i\}_{i=1}^N$ *define the reflection across the line* $\mathrm{span}(\mathbf{x}_0)$ *by* $R_{\mathbf{x}_0}(\mathbf{x}) := 2 \frac{\langle \mathbf{x}, \mathbf{x}_0 \rangle}{\|\mathbf{x}_0\|^2} \mathbf{x}_0 - \mathbf{x}$. *Assume the dataset is invariant under this reflection, i.e. for every index $i$ there exists an index $i'$ such that* $\mathbf{x}_0^{i'} = R_{\mathbf{x}_0}(\mathbf{x}_0^i)$. *Then for any $t \in (0, 1]$ and any* $\lambda \in \mathbb{R}$, $\mathbf{v}_t^{\mathrm{PF}}(\lambda \mathbf{x}_0) \in \mathrm{span}(\mathbf{x}_0)$ *(equivalently,* $\mathbf{v}_t^{\mathrm{PF}}(\lambda \mathbf{x}_0) \parallel \mathbf{x}_0$*). Consequently, the line* $\mathrm{span}(\mathbf{x}_0)$ *is invariant under the ODE* $\frac{d\mathbf{x}_t}{dt} = \mathbf{v}_t^{\mathrm{PF}}(\mathbf{x}_t)$, *and in particular the PF-ODE-Mean trajectory initialized at* $\mathbf{x}_\varepsilon = \alpha_\varepsilon \mathbf{x}_0$ *satisfies* $\mathbf{x}_t = \lambda(t) \mathbf{x}_0 \quad \forall t \in [\varepsilon, 1]$ *for a scalar function* $\lambda(t)$.

**Iterating *PF-ODE Mean*: *SVGD*-like refinement improves couplings.** Having fixed an unambiguous rule to construct couplings, we can now ask how to refine these couplings further to improve training. One empirical observation is that the first updates of *PF-ODE Mean* resemble those of *SVGD*, in the sense that both induce interactions between particles characterised by a Gaussian kernel. Concretely, consider the dynamics of $\mathbf{v}_t^{PF}$ under the initial conditions of *PF-ODE Mean*. Evaluating the closed-form vector field at this initialisation gives $\mathbf{v}_\epsilon^{PF}(\mathbf{x}_\epsilon^i) = a_\epsilon \mathbf{x}_0^i + b_\epsilon \sum_j w_{ij}(\epsilon) \mathbf{x}_0^j{}^\dagger$. In particular, $w_{ij}(\epsilon)$ are normalised Gaussian-kernel weights with $h_\epsilon = \frac{\sigma_\epsilon}{\alpha_\epsilon}$. Thus, early in time, *PF-ODE Mean* updates each particle via a kernel-weighted interaction with its neighbours, mirroring the kernel-interaction structure of SVGD updates (up to SVGD's additional repulsive term). Motivated by this connection, we hypothesise that running a few iterations of *PF-ODE Mean* can yield more regular, symmetric samples (the coupling is denoted by $\pi^{PF\,MeanIter}$). When running *PF-ODE Mean* for 50 iterations on the spiral dataset, we observe highly regular particle locations, reflecting variance-reduction behaviour rather than independent sampling (Figure 7a), and the resulting samples appear to reach fixed points (Figure 7b). Training on the couplings induced by this refined procedure yields substantially better performance than couplings obtained from *PF-ODE* and also improves over *SVGD*-based couplings (Figure 3).

**Voronoi-based semi-discrete couplings from *PF-ODE Mean* endpoints.** We found that *PF-ODE Mean* iterations can produce forward couplings that outperform standard *PF-ODE* couplings. Directly generating such couplings, however, is computationally infeasible in higher dimensions and on large datasets. Moreover, using a fixed set of forward couplings ties the number of available training pairs to the dataset size. This motivates an alternative use of the *PF-ODE Mean* endpoints. Empirically, the latent noise endpoints $\mathbf{x}_1$ produced by *PF-ODE Mean* tend to concentrate near the centres of latent clusters (Figure 4 C). We therefore can use a coupling $\pi^{(\cdot)}$ to induce a semi-discrete coupling $\hat{\pi}^{(\cdot)}$. Specifically, we sample $\mathbf{z} \sim \mathcal{N}(\mathbf{0}, \mathbf{I})$, assign it to its nearest neighbor $\mathbf{x}_1^i$ (i.e., $i = \arg\min_j \|\mathbf{z} - \mathbf{x}_1^j\|$), and train on the paired datapoint $(\mathbf{z}, \mathbf{x}_0^i)$. On the spiral dataset, Flow Matching trained with samples from $\hat{\pi}^{PF-Mean}$ significantly improves over standard Flow Matching (see Figure 3).

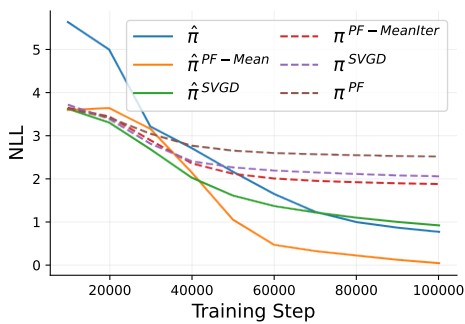

Figure 3: **Performance measured by NLL for different couplings on the 2D spiral dataset.** Dotted lines denote discrete coupling sets (one coupling per datapoint), and solid lines denote semi-discrete couplings.

---

$^\dagger a_\epsilon = \alpha_\epsilon \frac{d \log \sigma_t}{dt}\big|_{t=\epsilon}$, $b_\epsilon = \frac{d\alpha_t}{dt}\big|_{t=\epsilon} - \alpha_\epsilon \frac{d \log \sigma_t}{dt}\big|_{t=\epsilon}$, and $w_{ij}(\epsilon) := w_\epsilon^j(\alpha_\epsilon \mathbf{x}_0^i)$

Using samples from $\hat{\pi}^{PF-Mean}$ implicitly assumes that the relevant clusters in the prior are well captured by the Voronoi partition induced by the *PF-ODE Mean* endpoints $\{\mathbf{x}_1^i\}$. Empirically, the assignment induced by $\hat{\pi}^{(\cdot)}$ overlaps with the ground-truth coupling under $\mathbf{v}^{PF}$ to some extent, but the match is not exact. For example, on CIFAR-10, the accuracy of these naïve couplings drops to $\sim 33\%$, which is why we cannot carry the previous improvement to general data. To clarify when a Voronoi-based assignment is theoretically justified, we give sufficient symmetry conditions under which the dataset's Voronoi cells are invariant under the closed-form probability-flow field $\mathbf{v}^{PF}$ (Proposition 3.2, with detailed proof in Appendix C).

**Proposition 3.2.** *Let $\{\mathbf{x}_0^i\}_{i=1}^N \subset \mathbb{R}^d$ be a finite dataset such that $\|\mathbf{x}_0^i\| = \|\mathbf{x}_0^j\|$ for all $i, j$. Define Voronoi cells $V_i := \{\mathbf{x} : \|\mathbf{x} - \mathbf{x}_0^i\| \leq \|\mathbf{x} - \mathbf{x}_0^j\| \, \forall j\}$, and for $i \neq j$ define the bisector hyperplane*

$$H_{ij} := \left\{ \mathbf{x} : \|\mathbf{x} - \mathbf{x}_0^i\| = \|\mathbf{x} - \mathbf{x}_0^j\| \right\} = \{\mathbf{x} : \langle \mathbf{x}_0^j - \mathbf{x}_0^i, \mathbf{x} \rangle = 0\}. \tag{5}$$

*For a neighboring pair $(i, j)$ with $V_i \cap V_j \neq \emptyset$, let $n_{ij} := \mathbf{x}_0^j - \mathbf{x}_0^i$ and define the reflection across $H_{ij}$ by $R_{ij}(\mathbf{x}) := \mathbf{x} - 2\frac{\langle n_{ij}, \mathbf{x} \rangle}{\|n_{ij}\|^2} n_{ij}$. Assume that for every such neighboring pair $(i, j)$ the dataset is invariant under $R_{ij}$, i.e. for every index $k$ there exists an index $k'$ such that $\mathbf{x}_0^{k'} = R_{ij}(\mathbf{x}_0^k)$. Then for every such neighboring pair $(i, j)$, every $t \in (0, 1]$, and every $\mathbf{x} \in H_{ij}$, $\langle n_{ij}, \mathbf{v}_t^{PF}(\mathbf{x}) \rangle = 0$, so $\mathbf{v}_t^{PF}$ is tangent to $H_{ij}$. Consequently, each Voronoi cell $V_i$ is invariant under the flow of $\mathbf{v}_t^{PF}$.*

## 4  CONCLUSION

Closed-form probability–flow dynamics provide an analytic vector field $\mathbf{v}^{PF}$ along the Gaussian path, but they do not uniquely determine a one-to-one noise–data coupling. Empirically, naïvely instantiating $\mathbf{v}^{PF}$ as a finite set of endpoint pairs can yield ambiguous assignments and poor training on CIFAR-10, even though pretrained couplings appear compatible with at least one coupling realisable under $\mathbf{v}^{PF}$. On toy problems, enforcing non-ambiguous pairings improves training, and our symmetry results help to explain when structure is preserved. While our study is limited to toy settings, we hope that these results inspire future work on how to construct analytical (semi-)discrete couplings for training at scale.

ACKNOWLEDGMENTS

Part of this work was supported by the Helmholtz International Lab Causal Cell Dynamics (InterLabs-0029) through grant funding from the Initiative and Networking Fund of the Hermann von Helmholtz-Association Deutscher Forschungszentren e.V. The authors gratefully acknowledge the Gauss Centre for Supercomputing e.V. (www.gauss-centre.eu) for funding this project by providing computing time through the John von Neumann Institute for Computing (NIC) on the GCS supercomputer JUWELS (Jülich Supercomputing Centre, 2021) at the Jülich Supercomputing Centre (JSC). Furthermore, the authors acknowledge the computational resources provided by the National High Performance Computing Centre (www.nhr.kit.edu). TH acknowledges support from G-Research.

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

## DISCLOSURE OF LLM USAGE

Large language models were used for minor language editing, such as enhancing clarity, precision, and flow, and for aesthetic adjustments to figures to improve interpretability.

## REPRODUCIBILITY STATEMENT

All data used in this work are either publicly available or can be generated using the exact procedures we describe. In Appendix B, we provide detailed instructions for reproducing our experiments, along with references to the codebases on which our implementation is based.

## A   RELATED WORK

**Rectified flows.**   Rectified Flows methods (Liu et al., 2023) distil a pre-trained "teacher" into a faster "student" by constructing noise-data endpoint couplings from the teacher's probability-flow dynamics, and then training the student via Flow Matching (Lipman et al., 2023; Albergo & Vanden-Eijnden, 2023) on these induced pairs. By iterating this rectification procedure, sampling trajectories under the current model and retraining on the resulting couplings, the paths between endpoints get progressively straightened, enabling high-quality generation with few function evaluations. Recent work has shown that it is possible to achieve state-of-the-art results, with just one rectification procedure (Kim et al., 2025; Lee et al., 2024).

**Closed-form vector Fields.**   With an empirical data distribution and a Gaussian probability path, the associated probability-flow ODE admits an analytic (dataset-dependent) vector field. This connects to recent efforts that derive or analyse closed-form diffusion/flow constructions and what they imply about learning and generalisation (Bertrand et al., 2025; Scarvelis et al., 2025; Ryzhakov et al., 2024). Our work differs in focus. Rather than proposing a new closed-form model, we study how using the empirical probability-flow as a "teacher" affects the performance of few-step distilled students.

**Semi-discrete couplings and particle-based constructions.**   A central ingredient in training Flow models is the endpoint coupling between a discrete dataset and a continuous noise distribution. Tong et al. (2024); Pooladian et al. (2023) propose using minibatch-optimal transport to speed up training and sampling. Recent work makes this coupling choice explicit and develops semi-discrete constructions where one marginal is discrete and the other continuous, connecting Flow Matching to semi-discrete optimal transport and providing practical algorithms for coupling construction (Mousavi-Hosseini et al., 2025).

## B   EXPERIMENTAL DETAILS

In this section, we will briefly clarify the experimental settings we used. Specifically, we first give details on how we generate the closed-form couplings, then training details for CIFAR-10 and the Spiral data. While we have experimented using several functions for $\alpha_t$ and $\sigma_t$, we always observed the same behaviour. For all experiments presented in this paper, we use the standard form $\alpha_t = (1-t)$ and $\sigma_t = t$.

### B.1   CONSTRUCTING CLOSED-FORM COUPLINGS

**Integrating $\mathbf{v}^{PF}$**   Since we have only worked with relatively small datasets, we do not use any nearest neighbour approximation for Equation (3) as done in Scarvelis et al. (2025); Bertrand et al. (2025). Integrations were performed on 40GB A100 and 80GB H100 Nvidia GPUs. For integration, we used the `torchdiffeq` library (Chen, 2018), with the Runge-Kutta of order 5 of Dormand-Prince-Shampine method, setting the relative tolerance to `rtol` $= 10^{-8}$ and absolute tolerance to `atol` $= 10^{-10}$.

***SVGD*** To construct couplings using *SVGD*, we use the standard RBF-kernel, but adjust the bandwidth based on the median distance of the current datapoints. Specifically, we set the bandwidth at each sampling step $t$ to

$$h_t = \frac{\sqrt{\frac{1}{2}\texttt{median}\left(\{\|\mathbf{x}_t^i - \mathbf{x}_t^j\|^2\}_{i,j\in[1,...,N]}\right)}}{\log(N+1)}. \tag{6}$$

We use a stepsize of $\eta = 0.1$ and 10k sampling steps [‡].

### B.2 CIFAR-10 Experiments

For most of the setup, we follow Lee et al. (2024) [§]. However, since we need a fair comparison, we only use forward couplings for training, and therefore limit the amount of couplings to the dataset size (50k). Furthermore, we do not initialise our model with the EDM weights (Karras et al., 2022), but train from scratch without data augmentation. This again is done for a fair comparison, since training on augmented data would change the underlying ground truth vector field $\mathbf{v}^{PF}$. Models are trained for 200k steps, as we do not find any improvement in continuing training, due to the limited number of couplings.

### B.3 Spiral Dataset

The dataset consists of 200 samples, 100 for each spiral. The points are evenly spaced in $[0, \phi)$, with $\phi$ being the angular extent of each spiral. For the radius growth $r = (a\phi + b)$ we set the multiplier $a = 0.5$ and the offset to $b = \frac{1}{4}\pi$. For training, we use a 3-layer MLP with a hidden dimension of 256, the AdamW optimiser, and a learning rate of $10^{-4}$. All models are trained for 100k steps, and the negative log-likelihood is evaluated using 50 steps.

## C Proofs

In this section, we will provide the proofs for both our propositions. While both proofs can be combined, for clarity, we provide separate derivations for each proposition. We start by proving the radial invariance of *PF-ODE Mean*.

**Proposition 3.1.** *For each nonzero datapoint* $\mathbf{x}_0 \in \{\mathbf{x}_0^i\}_{i=1}^N$ *define the reflection across the line* $\mathrm{span}(\mathbf{x}_0)$ *by* $R_{\mathbf{x}_0}(\mathbf{x}) := 2\frac{\langle\mathbf{x},\mathbf{x}_0\rangle}{\|\mathbf{x}_0\|^2}\mathbf{x}_0 - \mathbf{x}$. *Assume the dataset is invariant under this reflection, i.e. for every index* $i$ *there exists an index* $i'$ *such that* $\mathbf{x}_0^{i'} = R_{\mathbf{x}_0}(\mathbf{x}_0^i)$. *Then for any* $t \in (0, 1]$ *and any* $\lambda \in \mathbb{R}$, $\mathbf{v}_t^{\mathrm{PF}}(\lambda\mathbf{x}_0) \in \mathrm{span}(\mathbf{x}_0)$ *(equivalently,* $\mathbf{v}_t^{\mathrm{PF}}(\lambda\mathbf{x}_0) \parallel \mathbf{x}_0$). *Consequently, the line* $\mathrm{span}(\mathbf{x}_0)$ *is invariant under the ODE* $\frac{d\mathbf{x}_t}{dt} = \mathbf{v}_t^{\mathrm{PF}}(\mathbf{x}_t)$, *and in particular the PF-ODE-Mean trajectory initialized at* $\mathbf{x}_\varepsilon = \alpha_\varepsilon\mathbf{x}_0$ *satisfies* $\mathbf{x}_t = \lambda(t)\mathbf{x}_0 \quad \forall t \in [\varepsilon, 1]$ *for a scalar function* $\lambda(t)$.

*Proof.* Recall the closed-form probability-flow field from Equation (3):

$$\mathbf{v}_t^{PF}(\mathbf{x}) = \sum_{i=1}^N w_t^i(\mathbf{x})\left(\frac{d\alpha_t}{dt}\mathbf{x}_0^i + \frac{d\log\sigma_t}{dt}\left(\mathbf{x} - \alpha_t\mathbf{x}_0^i\right)\right), \qquad w_t^i(\mathbf{x}) = \frac{\mathcal{N}(\mathbf{x} \mid \alpha_t\mathbf{x}_0^i, \sigma_t^2\mathbf{I})}{\sum_{j=1}^N \mathcal{N}(\mathbf{x} \mid \alpha_t\mathbf{x}_0^j, \sigma_t^2\mathbf{I})}. \tag{7}$$

Using $\sum_i w_t^i(\mathbf{x}) = 1$, we can rewrite this as

$$\mathbf{v}_t^{PF}(\mathbf{x}) = \frac{d\log\sigma_t}{dt}\mathbf{x} + \left(\frac{d\alpha_t}{dt} - \alpha_t\frac{d\log\sigma_t}{dt}\right)\sum_{i=1}^N w_t^i(\mathbf{x})\mathbf{x}_0^i. \tag{8}$$

Fix a nonzero $\mathbf{x}_0$ and define the reflection $R_{\mathbf{x}_0}(\mathbf{x}) = 2\frac{\langle\mathbf{x},\mathbf{x}_0\rangle}{\|\mathbf{x}_0\|^2}\mathbf{x}_0 - \mathbf{x}$. By assumption, for each index $i$ there exists an $i'$ with $\mathbf{x}_0^{i'} = R_{\mathbf{x}_0}(\mathbf{x}_0^i)$. Note, that from this it follows that $\|\mathbf{x}_0^i\| = \|\mathbf{x}_0^{i'}\|$.

---

[‡]For our implementation, we followed https://github.com/dilinwang820/Stein-Variational-Gradient-Descent/blob/master/python/svgd.py

[§]Their implementation is available under https://github.com/sangyun884/rfpp

Now take any $\mathbf{x} = \lambda\mathbf{x}_0$. Since $\mathbf{x}$ lies on the reflection axis, $R_{\mathbf{x}_0}(\mathbf{x}) = \mathbf{x}$. Because $R_{\mathbf{x}_0}$ is an isometry and $R_{\mathbf{x}_0}(\mathbf{x}) = \mathbf{x}$, we have

$$\|\mathbf{x} - \alpha_t \mathbf{x}_0^i\| = \|R_{\mathbf{x}_0}(\mathbf{x}) - \alpha_t R_{\mathbf{x}_0}(\mathbf{x}_0^i)\| = \|\mathbf{x} - \alpha_t \mathbf{x}_0^{i'}\|, \tag{9}$$

hence $\mathcal{N}(\mathbf{x} \mid \alpha_t\mathbf{x}_0^i, \sigma_t^2\mathbf{I}) = \mathcal{N}(\mathbf{x} \mid \alpha_t\mathbf{x}_0^{i'}, \sigma_t^2\mathbf{I})$ and therefore

$$w_t^i(\mathbf{x}) = w_t^{i'}(\mathbf{x}) \qquad \text{for all } \mathbf{x} = \lambda\mathbf{x}_0. \tag{10}$$

Consider the posterior mean term

$$m_t(\mathbf{x}) := \sum_{i=1}^{N} w_t^i(\mathbf{x})\,\mathbf{x}_0^i. \tag{11}$$

Group the dataset indices into reflection pairs $(i, i')$ (and possibly fixed points with $\mathbf{x}_0^i \parallel \mathbf{x}_0$, for which $R_{\mathbf{x}_0}(\mathbf{x}_0^i) = \mathbf{x}_0^i$). For each paired $(i, i')$, using Equation (10) we get

$$w_t^i(\mathbf{x})\mathbf{x}_0^i + w_t^{i'}(\mathbf{x})\mathbf{x}_0^{i'} = w_t^i(\mathbf{x})\big(\mathbf{x}_0^i + R_{\mathbf{x}_0}(\mathbf{x}_0^i)\big) = 2w_t^i(\mathbf{x})\,\text{proj}_{\mathbf{x}_0}(\mathbf{x}_0^i) \in \text{span}(\mathbf{x}_0), \tag{12}$$

and each fixed point $\mathbf{x}_0^i$ is already in $\text{span}(\mathbf{x}_0)$. Hence $m_t(\lambda\mathbf{x}_0) \in \text{span}(\mathbf{x}_0)$ for all $\lambda$ and all $t \in (0, 1]$.

Finally, Equation (8) shows that for $\mathbf{x} = \lambda\mathbf{x}_0$ both terms $\frac{d\log\sigma_t}{dt}\mathbf{x}$ and $\left(\frac{d\alpha_t}{dt} - \alpha_t\frac{d\log\sigma_t}{dt}\right)m_t(\mathbf{x})$ lie in $\text{span}(\mathbf{x}_0)$, so $\mathbf{v}_t^{PF}(\lambda\mathbf{x}_0) \in \text{span}(\mathbf{x}_0)$.

Since $\mathbf{v}_t^{PF}$ maps the line $\text{span}(\mathbf{x}_0)$ into itself for all $t \in [\varepsilon, 1]$, the line is an invariant set for the ODE. Therefore, if $\mathbf{x}_\varepsilon = \alpha_\varepsilon\mathbf{x}_0 \in \text{span}(\mathbf{x}_0)$, then $\mathbf{x}_t = \lambda(t)\mathbf{x}_0$ for all $t \in [\varepsilon, 1]$. $\qquad\square$

Next, we will move to the formation of invariant Voronoi cells under $\mathbf{v}^{PF}$. While we use very similar conditions as in the previous proof, we will restate those, including their implications

**Proposition 3.2.** *Let $\{\mathbf{x}_0^i\}_{i=1}^N \subset \mathbb{R}^d$ be a finite dataset such that $\|\mathbf{x}_0^i\| = \|\mathbf{x}_0^j\|$ for all $i, j$. Define Voronoi cells $V_i := \{\mathbf{x} : \|\mathbf{x} - \mathbf{x}_0^i\| \le \|\mathbf{x} - \mathbf{x}_0^j\| \; \forall j\}$, and for $i \ne j$ define the bisector hyperplane*

$$H_{ij} := \left\{\mathbf{x} : \|\mathbf{x} - \mathbf{x}_0^i\| = \|\mathbf{x} - \mathbf{x}_0^j\|\right\} = \{\mathbf{x} : \langle\mathbf{x}_0^j - \mathbf{x}_0^i, \mathbf{x}\rangle = 0\}. \tag{5}$$

*For a neighboring pair $(i, j)$ with $V_i \cap V_j \ne \emptyset$, let $n_{ij} := \mathbf{x}_0^j - \mathbf{x}_0^i$ and define the reflection across $H_{ij}$ by $R_{ij}(\mathbf{x}) := \mathbf{x} - 2\frac{\langle n_{ij}, \mathbf{x}\rangle}{\|n_{ij}\|^2}n_{ij}$. Assume that for every such neighboring pair $(i, j)$ the dataset is invariant under $R_{ij}$, i.e. for every index $k$ there exists an index $k'$ such that $\mathbf{x}_0^{k'} = R_{ij}(\mathbf{x}_0^k)$. Then for every such neighboring pair $(i, j)$, every $t \in (0, 1]$, and every $\mathbf{x} \in H_{ij}$, $\langle n_{ij}, \mathbf{v}_t^{PF}(\mathbf{x})\rangle = 0$, so $\mathbf{v}_t^{PF}$ is tangent to $H_{ij}$. Consequently, each Voronoi cell $V_i$ is invariant under the flow of $\mathbf{v}_t^{PF}$.*

*Proof.* Fix a neighboring pair $(i, j)$ with $V_i \cap V_j \ne \emptyset$ and write $n_{ij} := \mathbf{x}_0^j - \mathbf{x}_0^i$. Under the assumption $\|\mathbf{x}_0^i\| = \|\mathbf{x}_0^j\|$, the bisector hyperplane is

$$H_{ij} = \{\mathbf{x} : \|\mathbf{x} - \mathbf{x}_0^i\| = \|\mathbf{x} - \mathbf{x}_0^j\|\} = \{\mathbf{x} : \langle n_{ij}, \mathbf{x}\rangle = 0\}. \tag{13}$$

Using the same simplification as in Equation (8), we write

$$\mathbf{v}_t^{PF}(\mathbf{x}) = \frac{d\log\sigma_t}{dt}\mathbf{x} + \left(\frac{d\alpha_t}{dt} - \alpha_t\frac{d\log\sigma_t}{dt}\right)m_t(\mathbf{x}), \qquad m_t(\mathbf{x}) := \sum_{k=1}^{N} w_t^k(\mathbf{x})\,\mathbf{x}_0^k. \tag{14}$$

For $\mathbf{x} \in H_{ij}$, the first term satisfies $\langle n_{ij}, \frac{d\log\sigma_t}{dt}\mathbf{x}\rangle = \frac{d\log\sigma_t}{dt}\langle n_{ij}, \mathbf{x}\rangle = 0$, so it remains to show $\langle n_{ij}, m_t(\mathbf{x})\rangle = 0$.

Let $R_{ij}(\mathbf{x}) = \mathbf{x} - 2\frac{\langle n_{ij}, \mathbf{x}\rangle}{\|n_{ij}\|^2}n_{ij}$ be the reflection across $H_{ij}$. By assumption, for each $k$ there exists $k'$ with $\mathbf{x}_0^{k'} = R_{ij}(\mathbf{x}_0^k)$. Since $\mathbf{x} \in H_{ij}$ is fixed by the reflection ($R_{ij}(\mathbf{x}) = \mathbf{x}$) and $R_{ij}$ is an isometry, we have

$$\|\mathbf{x} - \alpha_t\mathbf{x}_0^k\| = \|R_{ij}(\mathbf{x}) - \alpha_t R_{ij}(\mathbf{x}_0^k)\| = \|\mathbf{x} - \alpha_t\mathbf{x}_0^{k'}\|, \tag{15}$$

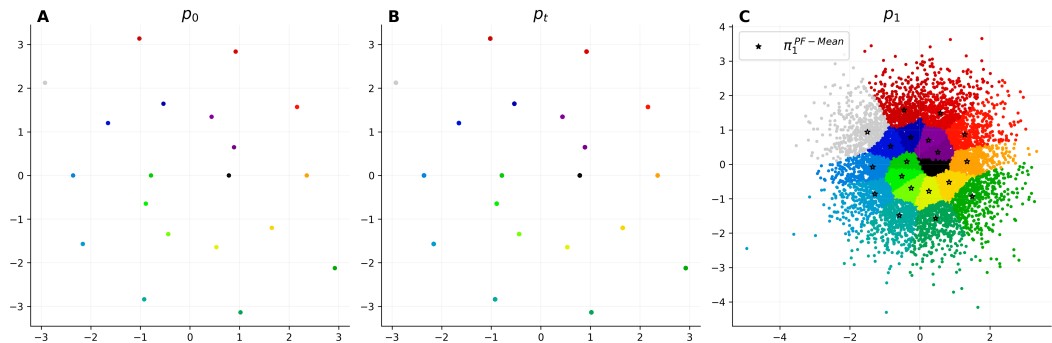

Figure 4: **Integrating *PF-ODE*.** The data distribution consists of two spirals (A), and we sample 500 points for each data point via $\alpha(t)\mathbf{x}_0 + \sigma(t)\mathbf{z}$, with $t = 10^{-4}$ and $\mathbf{z} \sim \mathcal{N}(\mathbf{0}, \mathbf{I})$ (B). The datapoints form clusters at $p_1$, meaning if integrated backwards, entire regions will contract to a single datapoint, while *PF-ODE Mean* seems to be centred in those clusters (C).

hence $w_t^k(\mathbf{x}) = w_t^{k'}(\mathbf{x})$ for all $\mathbf{x} \in H_{ij}$.

Moreover, by construction of the reflection,

$$\langle n_{ij}, \mathbf{x}_0^{k'} \rangle = \langle n_{ij}, R_{ij}(\mathbf{x}_0^k) \rangle = -\langle n_{ij}, \mathbf{x}_0^k \rangle. \tag{16}$$

Therefore, for each reflection pair $(k, k')$,

$$\langle n_{ij}, w_t^k(\mathbf{x})\mathbf{x}_0^k + w_t^{k'}(\mathbf{x})\mathbf{x}_0^{k'} \rangle = w_t^k(\mathbf{x})\big(\langle n_{ij}, \mathbf{x}_0^k \rangle + \langle n_{ij}, \mathbf{x}_0^{k'} \rangle\big) = 0. \tag{17}$$

Summing over all pairs (and noting points lying in $H_{ij}$ satisfy $\langle n_{ij}, \mathbf{x}_0^k \rangle = 0$ anyway) gives $\langle n_{ij}, m_t(\mathbf{x}) \rangle = 0$, hence $\langle n_{ij}, \mathbf{v}_t^{PF}(\mathbf{x}) \rangle = 0$ for all $\mathbf{x} \in H_{ij}$.

Finally, to conclude Voronoi invariance: define

$$f_{ij}(\mathbf{x}) := \|\mathbf{x} - \mathbf{x}_0^i\|^2 - \|\mathbf{x} - \mathbf{x}_0^j\|^2. \tag{18}$$

Under equal norms, $f_{ij}(\mathbf{x}) = 2\langle n_{ij}, \mathbf{x} \rangle$, so $H_{ij} = \{\mathbf{x} : f_{ij}(\mathbf{x}) = 0\}$ and

$$\frac{\mathrm{d}}{\mathrm{d}t} f_{ij}(\mathbf{x}_t) = \langle n_{ij}, \frac{\mathrm{d}\mathbf{x}_t}{\mathrm{d}t} \rangle = 2\langle n_{ij}, \mathbf{v}_t^{PF}(\mathbf{x}_t) \rangle. \tag{19}$$

Hence, whenever $\mathbf{x}_t \in H_{ij}$, we have $\frac{\mathrm{d}}{\mathrm{d}t} f_{ij}(\mathbf{x}_t) = 0$, so the flow is tangent to $H_{ij}$. Since each Voronoi cell $V_i$ is the intersection of spaces of the form $\{\mathbf{x} : f_{ij}(\mathbf{x}) \leq 0\}$ over its neighbouring $j$, trajectories cannot cross any boundary $H_{ij}$, and thus each $V_i$ is invariant under the flow. $\qquad\square$

## D   VISUALIZING $\mathbf{v}^{PF}$

To better understand the behaviour of the closed-form probability-flow field $\mathbf{v}^{PF}$ and the couplings it induces, we visualise the resulting dynamics across a range of settings. We begin with low-dimensional toy datasets, where the geometry of trajectories and contraction phenomena can be directly observed, and then extend our analysis to image data to illustrate how these effects persist and interact in high-dimensional spaces.

**Many-to-one contraction and cluster formation.**   Figure 4 illustrates the forward integration under $\mathbf{v}^{PF}$ starting from perturbed points $\mathbf{x}_\varepsilon = \alpha(\varepsilon)\mathbf{x}_0 + \sigma(\varepsilon)\mathbf{z}$, where $\mathbf{z} \sim \mathcal{N}(\mathbf{0}, \mathbf{I})$. When adding these small perturbations, multiple trajectories move into the same region at $t = 1$, forming clusters in the prior distribution. Interpreted as a coupling construction, this demonstrates the semi-discrete behaviour: large sets of initial conditions can map to the same datapoints when integrating backwards in time, so the resulting pairing is not canonically one-to-one.

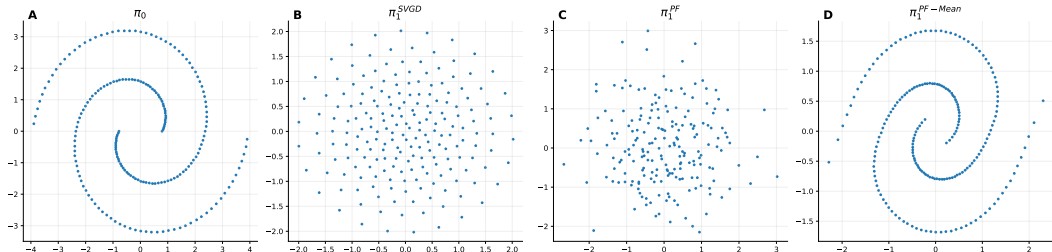

Figure 5: **Noise endpoints induced by different coupling constructions on a 2D spiral. A:** data samples $\mathbf{x}_0$. **B–D:** terminal noise endpoints $\mathbf{x}_1 \sim \pi_1^{(\cdot)}$ obtained by integrating the different closed-form fields. **B:** *SVGD* couplings yield well-spread endpoints. **C:** standard closed-form *PF-ODE* couplings (initialised with $\mathbf{x}_\epsilon = \alpha_\epsilon \mathbf{x}_0 + \sigma_\epsilon \mathbf{z}$ with $\epsilon = 10^{-4}$ and $\mathbf{z} \sim \mathcal{N}(\mathbf{0}, \mathbf{I})$) produce approximately Gaussian endpoints. **D:** the deterministic *PF-ODE Mean* initialization ($\mathbf{x}_\epsilon = \alpha_\epsilon \mathbf{x}_0$) preserves/warps the data geometry.

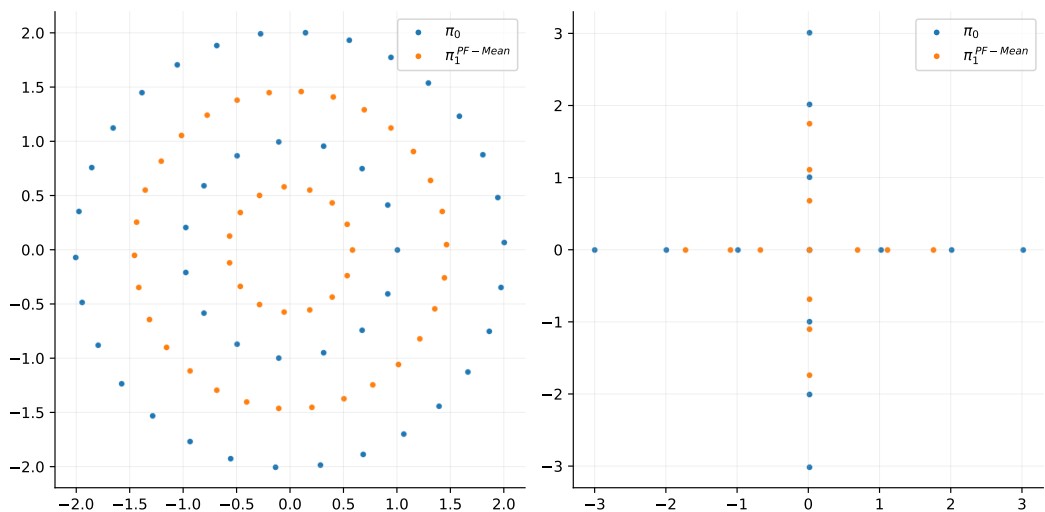

Figure 6: **Symmetry preservation under PF-ODE Mean on a circular toy dataset.** Starting from uniformly spaced points on a circle ($\pi_0$, blue), integrating the *PF-ODE Mean* dynamics maps samples to new locations ($\pi_1^{PF\ Mean}$, orange) while preserving the structure.

**PF-ODE Mean.** Figure 2 compares standard *PF-ODE* coupling generation (randomized $\mathbf{x}_\varepsilon = \alpha(\varepsilon)\mathbf{x}_0 + \sigma(\varepsilon)\mathbf{z}$) to the deterministic initialisation used by *PF-ODE Mean* (setting $\mathbf{x}_\varepsilon = \alpha(\varepsilon)\mathbf{x}_0$). Note that this is not the exact mean of the marginal, but the mean of the conditional at time $\epsilon$. Although these initialisations differ only by a minimal perturbation, the resulting endpoints differ enormously. In particular, randomised initialisation yields endpoints that resemble Gaussian noise, whereas *PF-ODE Mean* retains the structure so well that one can identify couplings visually.

**Comparing couplings across methods.** Figure 5 shows the difference of $\pi_1^{(\cdot)}$ induced by different constructions on a low-dimensional dataset. Endpoints generated via *SVGD* tend to be well spread, consistent with the combination of attraction toward high-density regions and repulsive interactions that promote coverage of the target distribution. In comparison, *PF-ODE* endpoints are approximately Gaussian overall but can exhibit local aggregation patterns. Finally, *PF-ODE Mean* endpoints preserve substantial aspects of the original data geometry, highlighting that a deterministic $\varepsilon$-start can bias the flow toward structure-preserving trajectories rather than fully mixing into the Gaussian target.

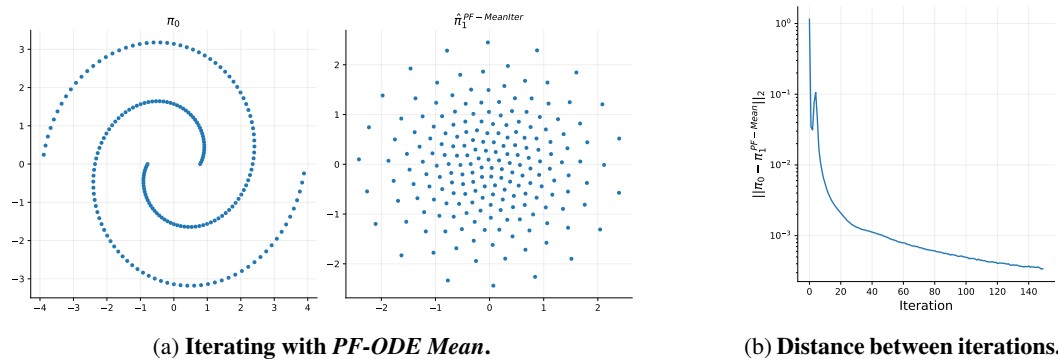

(a) **Iterating with *PF-ODE Mean*.**          (b) **Distance between iterations.**

Figure 7: **Approaching empirical fixed points when iterating over *PF-ODE Mean*.** (a) When applying *PF-ODE Mean* iteratively to a dataset, we observe that the initial structure gets washed out and the corresponding samples resemble samples from *SVGD*. (b) The resulting samples appear to be fixed points of the iterative procedure.

**Symmetry preservation under *PF-ODE Mean*.** Figure 6 visualises a symmetric data set that satisfies the conditions of Proposition 3.1. We observe that *PF-ODE Mean* only scales datapoints while preserving the global structure.

**Iterating *PF-ODE Mean*** Figure 7 shows the effect of iterating *PF-ODE Mean*. Empirically, repeated application progressively reduces the visible structure of the point cloud and yields increasingly regular configurations, while the distance between successive iterates decays, suggesting convergence toward empirical fixed points. Although such iterations are not intended as a practical coupling mechanism in high dimensions, they can improve performance in training.

