# OpenReview forum: "On Closed-Form Couplings"
_ICLR.cc/2026/Workshop/GRaM — ICLR 2026 Workshop GRaM Poster_

### Official Review · Reviewer_sNWP · 2026-02-09

**Rating:** 7
**Confidence:** 3

**Review:**

For a finite dataset and a Gaussian probability path one can write down the coupling in closed form.
The authors study several properties of these closed form couplings. In my opinion the topic is a good fit for the workshop. I like the experiments on the 2D spiral, they offer a good intuition for the results.

I have a few questions & comments:
1. The authors state in the abstract "Surprisingly, these [closed-form] couplings turn out to be poor teachers and
significantly reduce the performance of the student.". However, in [1] the authors explicitly show that the closed-form coupling can be a good teacher (see Section 4.1. Empirical flow matching in [1]). Meaning that regressing against the optimal velocity arising from the closed-form coupling actually achieves a (slight) superior performance over conditional flow matching. Maybe the authors can specify their claim?
2. Was the velocity $v^{FM}$ obtained via the usual conditional flow matching loss or using regression against $v^{PF}$ in Equation (2) (i.e., Empirical Flow Matching in Bertrand et al. (2025))?
3. You start integrating with $v^{PF}$ at $t=\epsilon$. How robust are the results against the choice of $\epsilon$?
4. What is the different between the couplings $\hat{\pi}$ and $\hat{\pi}^\text{PF-ODE}$?
5. You show that integrating from $x_\epsilon = \alpha_\epsilon x_0$ using PF-ODE velocity does not lead to a Gaussian distribution at $t=1$. Does integrating using $v^\text{FM}$ (the flow matching velocity) lead to a Gaussian distribution?

References:
- [1] Bertrand et al. (2025): "On the Closed-Form of Flow Matching: Generalization Does Not Arise from Target Stochasticity" (NeurIPS)

**Pmlr Suitability:**

NA

---

### Official Review · Reviewer_71GW · 2026-02-23

**Rating:** 7
**Confidence:** 3

**Review:**

Pros:

* The paper investigates an interesting and (initially) counter-intuitive result that closed-form couplings are not performant for distillation via rectified flows, despite being the "optimal" solution to the flow matching objective and the fact pre-trained couplings are empirically consistent with the closed-form coupling.
* The paper provides a nice analysis of exactly why closed-form couplings fail and provides good empirical evaluations for testing the properties of closed-form couplings and extensions for improving the performance of couplings using the closed-form vector field (e.g. they propose and look at PF-ODE Mean, SVGD couplings etc.)
* The paper is nicely presented and written.

Cons:

* It would have been nice to provide (little) more background on rectified flows in the body of the paper due their importance to the paper. For example, how the sampled coupling is then used to train the distilled model.


Typos:

* Line 73: "\mathbf{z}.." (repeated full stops)

**Pmlr Suitability:**

NA

---

### Official Review · Reviewer_bT2v · 2026-02-24

**Rating:** 7
**Confidence:** 2

**Review:**

The work presents a surprising result that closed form couplings serves as a poor teacher for distilling a few-step "student" flow. Phenomena is analyzed both empirically and theoretically.

Pros
- Authors present a very interesting, non-intuitive result and consequently analyze reasons behind the observed phenomena.
- Strong empirical and theoretical analysis of subsequent research questions.

Cons
- The paper would benefit from providing more context in writing on why this problem is important for the field.
- As the paper focus on rectified flows, it would be good to have at least some mention and explanation of it in background section.
- In paragraph "Smooth one-to-one couplings via Stein Variational Gradient Descent (SVGD)" - Figure 2 results could be better commented, e.g., what we compare to what as Figure 2 aggregate multiple results.

Suggestions
- Title could be a bit more elaborative to better convey the paper essence.
- Given information density of the paper, the overall results would benefit from longer form publication.

**Pmlr Suitability:**

NA

---

### Meta-Review · Area_Chair_RHRB · 2026-02-25

**Decision:**

Accept

**Metareview:**

The paper has received favourable reviews. All reviewers agree that the paper introduces an interesting and counterintuitive result with strong theoretical and empirical analysis, with only minor weaknesses mentioned. Therefore I recommend accepting the paper.

**Relevance To Proceedings:**

Tiny paper — does not apply

**Relevance To Workshop:**

Yes — suitable for GRaM

---

### Decision · Program_Chairs · 2026-03-02

Accept (Poster)